# Exploring the Impact of a Structured Educational Approach on Peristomal Skin Complications: An Interim Analysis

**DOI:** 10.3390/healthcare12181805

**Published:** 2024-09-10

**Authors:** Francesco Carlo Denti, Eliana Guerra, Francesca Caroppo, Pietro Abruzzese, Fabrizio Alessi, Filippo Barone, Pasqualina Bernardino, Massimiliano Bergamini, Cristina Bernardo, Gloria Bosio, Paula Carp, Manuela Cecconello, Annalinda Cerchier, Francesca Croci, Rita Detti, Cristina Di Pasquale, Maria Rosaria D’Ippolito, Simona Ditta, Erica Ducci, Anna Belloni Fortina, Stefano Frascarelli, Marianna Galante, Rita Guarino, Nicola Leggio, Elisabetta Livio, Alessandra Marchetti, Francesca Marelli, Rita Mastropaolo, Viviana Melis, Nicola Palmiero, Arianna Panarelli, Anna Lea Pascali, Francesco Pizzarelli, Laura Precisi, Cinzia Rastello, Silvia Regaglia, Rossana Elvira Rinaldi, Nadia Rumbolo, Claudio Sansone, Angela Santelli, Giovanni Sarritzu, Stefano Sfondrini, Sara Stanzani, Mattia Stella, Margherita Walterova, Rosario Caruso

**Affiliations:** 1Stomacare Service, IRCCS San Raffaele Institute, 20132 Milan, Italy; 2Enterostomal Rehabilitation Clinic, ASST Spedali Civili Brescia, 25123 Brescia, Italy; 3Department of Medicine DIMED, University of Padua, 35131 Padua, Italy; francesca.caroppo@outlook.it; 4Stomacare Service, Ospedale Bellaria Carlo Alberto Pizzardi, 40139 Bologna, Italy; 5Stomacare Service, Ospedale di Legnano—ASST Ovest Milanese, 20025 Legnano, Italy; 6Stomacare Service, Ospedale Generale Regionale Francesco Miulli, 70021 Acquaviva delle Fonti, Italy; 7Stomacare Service, Ospedali Riuniti Villa Sofia, 90146 Palermo, Italy; 8Stomacare Service, Presidio Ospedaliero Universitario “Santa Maria della Misericordia”, 33100 Udine, Italy; 9Stomacare Service, Presidio Ospedaliero di Ivrea—ASL Torino 4, 10015 Ivrea, Italy; bmc1272@gmail.com; 10Stomacare Service, Ospedale di Rivoli, 10098 Rivoli, Italy; gloria.bosio65@gmail.com; 11Stomacare Service, Presidio Ospedaliero Martini, 10141 Torino, Italy; 12Stomacare Service, Ospedale di Feltre, 32032 Feltre, Italy; 13Stomacare Service, Ospedale di San Donà di Piave, 30027 San Donà di Piave, Italy; 14Stomacare Service, Ospedale “Val Vibrata” di Sant’Omero—ASL Teramo, 64027 Teramo, Italy; 15Stomacare Service, Azienda Ospedaliero-Universitaria Senese, 53100 Siena, Italy; r.detti@ao-siena.toscana.it (R.D.);; 16Stomal Therapy Outpatient Service, European Institute of Oncology IRCCS, 20141 Milan, Italy; 17Stomacare Service, Azienda Ospedaliera di Rilievo Nazionale Antonio Cardarelli, 80131 Napoli, Italy; 18Stomacare Service, Ospedale Luigi Sacco, 20157 Milan, Italy; 19Department of Womens’ and Children’s Health (SDB), University of Padova, 35131 Padua, Italy; 20Stomacare Service, USL Umbria 1, 06127 Perugia, Italy; 21Stomacare Service, Ospedale Santa Chiara, 38123 Trento, Italy; marianna.galante@apss.tn.it; 22Stomacare Service, IRCCS Fondazione G. Pascale di Napoli, 80131 Napoli, Italy; 23Stomacare Service, Fondazione IRCCS Ca’ Granda Ospedale Maggiore Policlinico, 20122 Milan, Italy; 24Stomacare Service, Clinica Ospedaliero-Universitaria Policlinico Umberto I, 00161 Roma, Italy; 25Stomacare Service, Fondazione IRCCS Istituto Nazionale dei Tumori, 20133 Milan, Italy; 26Stomacare Service, Ospedale di Bolzano, 39100 Bolzano, Italy; 27Stomacare Service, ASST Papa Giovanni XXIII, 24127 Bergamo, Italy; 28Stomacare Service, Policlinico di Bari Ospedale “Giovanni XXIII”, 70124 Bari, Italy; 29Stomacare Service, Asl Lecce, 73100 Lecce, Italy; 30Stomacare Service, Azienda Ospedaliera di Padova, 35131 Padua, Italy; 31Stomacare Service, Azienda Ospedaliero Universitaria Pisana, 56126 Pisa, Italy; 32Stomacare Service, Azienda Ospedaliera Universitaria San Luigi Gonzaga, 10043 Orbassano, Italy; 33Stomacare Service, Ospedale Civile Santissima Annunziata, 07100 Sassari, Italy; 34Stomacare Service, Ospedale Pesenti Fenaroli, 24022 Alzano Lombardo, Italy; 35Stomacare Service, Fondazione IRCCS San Gerardo dei Tintori, 20900 Monza, Italy; 36Stomacare Service, Azienda Ospedaliera San Giovanni Addolorata, 00184 Roma, Italy; 37Stomacare Service, Ospedale Pio XI, 20832 Desio, Italy; 38Stomacare Service, Policlinico Universitario Monserrato “Duilio Casula”, 09042 Monserrato, Italy; 39Stomacare Service, Ospedale S. Anna—ASST Lariana, 22042 Como, Italy; stefano.sfondrini@asst-lariana.it; 40Stomacare Service, Ospedale Santa Maria delle Croci, 48121 Ravenna, Italy; 41Stomacare Service, IRCCS Saverio De Bellis, 70013 Castellana Grotte, Italy; 42Health Professions Research and Development Unit, IRCCS Policlinico San Donato, 20097 San Donato Milanese, Italy; 43Department of Biomedical Sciences for Health, University of Milan, 20133 Milan, Italy

**Keywords:** ostomy care, peristomal skin complications, patient stratification, educational intervention, multiple correspondence analysis, Poisson regression, specialized nurses

## Abstract

This study, employing an interim analysis, investigates the effects of the Dermamecum protocol, a structured educational and tailored approach that stratifies ostomy patients into risk paths (green, yellow, red) based on pre-operative and post-operative characteristics. The green path indicates a low risk of peristomal skin complications (PSCs), focusing on sustaining healthy behaviours and basic stoma care. The yellow path represents a moderate risk, emphasizing the need for patients to self-monitor and recognize early signs of complications. The red path corresponds to high risk, requiring stringent monitoring and immediate access to healthcare support. The study aims to reduce PSCs and improve patient outcomes. Methods include the stratification of 226 patients, with significant differences in gender distribution, BMI categories, and stoma types across the paths. Results show an occurrence rate of PSCs of 5.9% in all risk paths (5.7% green path, 4.7% yellow path, and 7.9% red path, *p* = 0.685), significantly lower than the median rate of 35% reported in the literature. Multiple correspondence analysis validated the stratification, with distinct clusters for each path. Poisson regression models in the exploratory framework of an interim analysis identified male gender as the only significant predictor of PSCs, indicating the need for gender-specific interventions. The findings suggest that the Dermamecum protocol effectively reduces early PSCs, providing a foundation for further research.

## 1. Introduction

Ostomies are the result of a surgical procedure where an opening is created on the front of the abdomen to expel waste, such as urine and/or faeces [1]. This condition is prevalent worldwide, with an estimated number of approximately 700,000 people in Europe living with an ostomy [1,2]. The conditions that typically necessitate an ostomy include cancer, inflammatory bowel diseases, trauma, and familial adenomatous polyposis [3].

Patients with a stoma are at considerable risk of developing peristomal skin complications (PSCs), which could significantly impact their quality of life [4,5,6,7,8]. PSCs encompass a range of issues, such as dermatitis, pruritis/xerosis, infections, and ulcerations [7]. These complications arise primarily due to the continuous exposure of the peristomal skin to effluent, which contains digestive enzymes, bile salts, and bacteria that can irritate the skin [4,5,6,7,8]. The compromised skin barrier function in stoma patients makes the skin more susceptible to inflammation and infection, leading to the development of conditions like dermatitis and ulcerations [5]. Additionally, factors such as mechanical trauma from repeated appliance changes and pressure from the stoma appliance could exacerbate these conditions. These pathophysiological mechanisms present a substantial challenge for patients and healthcare providers, often leading to increased healthcare utilization, discomfort, and reduced quality of life [5]. The literature highlights the prevalence and severity of PSCs among patients with ostomies. Studies have shown that up to 60% of patients with a stoma experience some form of skin complication [1,5].

Common complications include dermatitis and pruritis/xerosis, with psoriasis and obesity being significant risk factors. Additionally, infections and ulcerations are associated with factors such as atopic dermatitis, being underweight, and receiving radiotherapy [3]. A systematic review and meta-analysis further underscored the high incidence of stomal complications, which ranged from 9% to 63%, with a median of 35% in the first month after surgery, highlighting the need for targeted interventions [1].

Given the substantial impact of PSCs, their effective management and prevention require a structured approach tailored to patients’ risk profiles. This stratified approach is essential to address the varying risk levels and to implement appropriate preventive measures and treatments, as highlighted by recent research [7]. Effective management and prevention of PSCs necessitate a tailored approach based on individual risk profiles. Stratifying patients by their risk of developing PSCs could allow healthcare providers to deliver personalized care that can improve patient outcomes [9]. This stratification is crucial for implementing appropriate interventions matched to each patient group’s specific needs and risks [7].

Despite the known prevalence and impact of PSCs, there is a pressing need for validated strategies to stratify patients by risk and effectively implement tailored interventions [10]. The involvement of specialized nurses who lead interprofessional educational actions, including collaboration with dermatologists, surgeons, urologists, and other healthcare professionals, is crucial. Such experiences are needed in the literature to allow researchers to develop and validate effective, tailored interventions to prevent PSCs. For this reason, in the context of an interim analysis, this study aims to describe the overall complication rates among patients with an ostomy following a structured educational approach based on stratifying patients by their risk of peristomal complications. Based on previous evidence [7], the educational approach involves categorizing patients into different risk paths and providing specific educational and follow-up protocols tailored to their risk levels. Additionally, this interim analysis seeks to compare the characteristics of patients within each risk path to ascertain their distinct features, validate the discriminant capacity of the stratified approach to ensure accurate patient categorization, and identify predictors of complications through an interim exploratory analysis to refine and enhance future stoma care practices.

## 2. Materials and Methods

### 2.1. Design

This study is a cross-sectional study and represents an interim analysis. An interim analysis is a type of statistical analysis that is conducted at one or more points during a research study before the final data collection is complete. Its purpose is to evaluate the data that have been gathered up to that point to make early assessments about the effects and safety of the research. Interim analyses help in making decisions about continuing, modifying, or stopping a study earlier based on predefined criteria, such as the achievement of enough power to achieve the primary endpoint of a clinical study. In this study, the interim analysis is justified by the need for enough preliminary data, as required by the study protocol, to allow an iterative approach in determining the final sample size for the main study and explore the initial effect of the structured educational approach on PSCs with an explorative purpose. The main study aims to determine the effects of the new educational approach on PSCs. Gathering interim data enables the researchers to assess the initial effects of the risk stratification-based educational approach, make necessary adjustments, and optimize the educational protocols and follow-up procedures. This approach ensures that the final study design is robust and adequately powered to detect the significant effects of the intervention. The reporting of this study is consistent with the Strengthening the Reporting of Observational Studies in Epidemiology (STROBE) guideline [11].

### 2.2. The Risk Stratification-Based Educational Approach

The educational approach implemented in this study is based on stratifying patients by their risk of developing PSCs. The authors call this approach Dermamecum. The Dermamecum approach involves categorizing patients into three distinct risk paths during the first visit with the specialist nurse after patients’ hospital discharge (generally two weeks after surgery)—green, yellow, and red—each tailored to the specific risk level of the patient based on a previous study [7]. This approach aims to provide tailored education and support that aligns with each patient’s specific risk level, promoting effective self-care, timely intervention, and improved patient outcomes.

#### 2.2.1. Stratification Process and Risk Calculation

Patients are stratified into one of the three risk paths based on a comprehensive assessment of their demographic, clinical, and post-operative characteristics, which are used to calculate the percentage risk for each type of PSC (i.e., dermatitis, itchiness, infections, and ulcers) [7]. The overall risk percentage, which determines the patient’s risk path, is derived from these individual risk percentages. This methodology allows for a precise categorization of patients into green, yellow, or red paths based on their calculated risk of PSCs, as described in Table 1.

The risk of dermatitis is influenced by factors such as the presence of psoriasis, chemotherapy treatment, the use of one-piece ostomy devices, and an irregular stoma profile, while protective factors include the use of hydrocolloid plaques, low pH detergents, and a low-residue diet [7]. The risk of itchiness (pruritis/xerosis) is higher for those who are overweight or obese, male, or using deep convex plaques, whereas older age and a sedentary lifestyle are protective factors [7]. Infections are more likely in patients with Class 2 obesity, frequent physical activity, or hernias, but the use of one-piece devices can reduce this risk [7]. Ulceration risk is elevated in underweight individuals, those who are non-autonomous, or those with inflammatory bowel disease (IBD) or a history of radiotherapy, while protective factors include the use of protective film and a regular abdominal stoma profile. Based on these individual risk percentages, an overall risk percentage is calculated for each patient, which allows us to assign patients to a specific path.

#### 2.2.2. Characteristics of Each Path in the Dermamecum

The green path is for patients with an overall risk of lower than 25% of PSCs. The objective for these patients is to focus on sustaining healthy behaviours and proper stoma care techniques to maintain their current health status [12]. Based on the literature [12,13,14], the educational content for this group includes basic self-care maintenance. This includes instructions on proper stoma cleaning, correct appliance fitting, and routine care practices aimed at maintaining their current health status. The educational materials are designed to be straightforward and are based on best practices for daily stoma care.

The yellow path is for patients with an overall risk of PSCs between 25% and 75%. The objective for these patients is to strengthen their knowledge regarding the need to self-monitor changes in peristomal skin and react in a timely manner to any changes [12,13,14]. This path includes all elements of the green path, with additional information on recognizing early signs of complications, such as dermatitis and infections, and guidance on when to seek medical advice. In other words, this path includes an enhancement focused on self-monitoring peristomal skin, identifying abnormal changes, and understanding when to seek medical advice. This approach focuses on empowering patients to manage their care more proactively by recognizing and addressing potential issues before they escalate.

The red path is for patients with an overall risk of higher than 75% of PSCs. The objective for these patients is to implement more stringent monitoring and provide them with additional options to contact healthcare providers promptly when signs or symptoms of complications are noted. This path includes all elements of the green and yellow paths, frequent follow-up appointments in person or via phone, and direct lines of communication with stoma care specialists for immediate assistance. Therefore, patients are educated on more advanced warning signs of complications and are provided with clear instructions on how to contact healthcare providers promptly. The focus is on preventing severe complications through close monitoring and rapid response to any signs of PSCs.

The Dermamecum approach ensures that each patient receives education and support tailored to their specific risk level, promoting effective self-care and timely intervention to prevent and manage PSCs. The approach aims to improve patient outcomes and reduce the incidence and severity of PSCs by stratifying patients and customizing their educational content in order to trigger best behaviours when maintaining peristomal health, monitoring changes, and making decisions when they encounter any change [12].

### 2.3. Context and Procedure

The context of this study is a network of Italian stoma care services called the Skin Health Academy, created in early 2020. In this network, specialized stoma care nurses have standardized their proactive educational approach to reduce PSCs. Whenever possible, these nurses visit patients before surgery for stoma siting and begin building a relationship with the patients. The Dermamecum resulted from 3 years of networking and educational initiatives and was operational in 2023. Standard education is delivered during the post-operative phase, and the Dermamecum program starts two weeks after surgery. Future revisions of the approach may move the start of Dermamecum to an earlier phase, but the current timing was chosen to ensure that all nurses in the network could consistently implement the program. In this phase, the Dermamecum program lasts up to three months after surgery.

This interim study focuses only on early-onset complications, defined as those observed within 30 days of the operation. As a preventive approach, Dermamecum is intended for patients who do not have PSCs at the start of the program.

In the context of this study, the terms “educational approach”, “educational initiative”, and “standard education” have distinct meanings. The “educational approach” refers to the overarching methodology used to deliver patient education. In this study, it is represented by the Dermamecum. This structured, risk-stratification-based system tailors educational content and support according to each patient’s specific risk of peristomal skin complications. This approach is systematic and designed to align educational efforts with the patients’ unique needs and risk profiles. “Educational initiative” describes specific programs or interventions implemented within the broader educational approach. These initiatives might include targeted campaigns, workshops, or specialized materials developed to address particular needs identified within the patient population. For instance, a specific initiative might be a series of educational sessions or materials focusing on the early detection of complications for patients in the yellow path. On the other hand, “standard education” refers to the general education that all patients typically receive as part of routine care. It is usually more generic, covering basic stoma care and hygiene without consideration of individual risk factors. Standard education serves as the foundational knowledge provided to all patients, regardless of their risk stratification.

### 2.4. Measures

The study measured various demographic, clinical, and post-operative variables to assess the characteristics and outcomes of patients in the different risk paths, assessed as routine clinical practice by nurses involved in the Skin Health Academy network. These measures included sex, recorded as male, female, or other, and age, recorded in years. Body mass index (BMI) was categorized as underweight, normal, overweight, or obese. Comorbidities such as diabetes, IBD, hematologic disorders, cardiovascular disorders, psoriasis, atopic dermatitis, and allergic dermatitis were documented in the case report forms from clinical records.

Previous treatments were noted, including whether patients had undergone neoadjuvant chemotherapy or radiotherapy and whether they had any prior surgical procedures. The type of stoma planned was classified as colostomy, ileostomy, or urostomy, and whether a stoma siting was scheduled before surgery was recorded.

Job types were classified as sedentary, semi-sedentary, non-sedentary, or not working, and sports activity frequency was categorized as never, rarely, often, or always. Baseline risks were assessed, including percentage risk of dermatitis, itchiness, infections, ulcers, and average risk of complications, with median and interquartile range.

Post-operative measures included whether the scheduled stoma site was implemented, and dietary habits were categorized as having a low residue for ileostomy, having no special diet, or other. The abdomen profile was described as regular, retroflexed, or introflexed, and the physical form of the stoma post-surgery was classified as extra-flexed, normal, low, flat, or introflexed. Post-surgical herniation and prolapse complications were recorded, including incidences of incisional hernia, prolapse, and hernia.

The type of device used was recorded, whether single-piece or two-piece, and the type of baseplate used was categorized as flat, traditional convex, or deep convex. The baseplate material was recorded as hydrocolloid or hydrocolloid plus non-woven textile. Caregiver support was categorized as full, partial, supervisory, or none.

The use of various ostomy care products was documented, including paste with alcohol, alcohol-free paste, protective film, remover, belt, rings, powder, extenders, neutral pH soap, acid pH soap, basic pH soap, non-woven gauze, and wet wipes. Early complications observed within 30 days were documented and represented the primary outcome for this study, noting also whether they were prevented based on baseline risks.

### 2.5. Eligibility Criteria to the Dermamecum

All patients aged 18 and above who underwent a surgical procedure resulting in the creation of a colostomy, ileostomy, or urostomy were eligible for this study. To ensure proper follow-up and intervention, patients needed to be under the care of a specialized nurse in stoma care affiliated with the Skin Health Academy. Exclusion criteria included the presence of a PSC before initiating the Dermamecum protocol, cognitive impairment, and other conditions that could interfere with the ability to participate in the educational program.

### 2.6. Ethical Considerations

This study adhered to the ethical principles outlined in the Declaration of Helsinki. The research protocol, including the need for an interim analysis, was reviewed and approved by the institutional review board of the Skin Health Academy in Italy (protocol n. 1/int/2023). The study complied with all applicable good clinical practices of the International Council for Harmonisation of Technical Requirements for Pharmaceuticals for Human Use. Given the nature of the data collection, which involved routine data gathered by specialized nurses as part of standard care, study-specific written patient consent was waived, while the consent used in the investigation settings was assessed to be adequate to inform patients regarding the clinical activities encompassed in this protocol. Data collection was carefully designed to ensure the anonymity and confidentiality of all patient data. The study design also ensured that no participants were exposed to unnecessary risks, and all procedures were implemented with the utmost care to maintain the integrity and ethical standards of the research.

### 2.7. Data Analysis

Data were checked for their distribution and missingness, with any missing data managed using the last observation carried forward method in relation to the outcome (only one patient had missing data). The information was summarised based on the variables’ nature and distribution. Inferential comparisons between the paths (green, yellow, and red) were performed, considering the distribution and the nature of the metric of the original variable. Statistical significance was adjusted for multiple comparisons using the Benjamini–Hochberg procedure to control the inflation on type I error of the multiple comparisons [15].

To explore the predictors of PSCs, we employed a multi-step approach involving data cleaning, multiple correspondence analysis (MCA), and Poisson regression modelling [16,17].

First, we defined a set of predictors based on the literature [7]. These predictors included various demographic, clinical, and treatment-related variables, as follows: presence of psoriasis, chemotherapy, type of device used, type of plate used, use of acidic pH soap, use of non-woven gauze, diet type, gender, BMI category, use of convex plate, use of protective film, caregiver support mode, presence of hernia, sport frequency, presence of inflammatory bowel disease, radiotherapy, and the risk stratification path [7]. The outcome variable was created by summing the values of specific complication-related columns regarding 30-day assessments from the surgery, as follows: dermatitis, infections, itching, and ulcerations. Only one patient had missing data and this was handled using the last observation carried forward method (no complications were carried out from the baseline assessment). We then selected the relevant columns, comprising both predictors and the outcome variable, and removed rows with missing values in these columns.

Second, the predictors identified by the literature were subjected to MCA to reduce dimensionality and identify the principal components that capture the variance in the predictors [18,19]. The MCA results were visualized using scree plots and biplots to understand the distribution and relationships of the predictors [20].

The principal components derived from the MCA explaining more than 6% of variance were then used as predictors in a Poisson regression model, with the outcome variable being the sum of the PSCs per patient. As shown in the equation below, we included interaction terms between the risk stratification paths and the principal components to capture any differential effects across the paths. The gender variable was kept outside of the MCA components because it was not summarized in the MCA.
(1)$$\log\left(E\left[Y_{i}\right]\right)=\beta_{0}+\beta_{\mathrm{gender}}{\mathrm{Gender}}_{i}+\sum_{j=1}^{k}\beta_{j}PC_{\mathrm{ij}}+\sum_{j=1}^{k}\sum_{l=1}^{m}\gamma_{jl}\left(PC_{\mathrm{ij}}\times \mathrm{Pat}h_{i}\right)$$

In this model, E[Y_i_] is the expected value of the outcome for a patient “i”, β_0_ is the intercept term, β_gender_ is the coefficient for gender in patient “i”, β_j_ are the coefficients for the *j*-th principal components (PC_ij_) referring to patient “i”, and γ_jl_ are the coefficients for the interaction terms between principal components and risk stratification (Path_i_).

The initial and stepwise Poisson regression models were summarized, and their coefficients (incidence rate ratio, IRR), 95% confidence interval (CI), and *p*-values were reported. Additionally, model fit statistics such as the Akaike information criterion (AIC) and deviance were calculated to compare the models’ performance [21]. The stepwise procedure involved iteratively adding or removing predictors based on their statistical significance to optimize the model fit. Starting with the initial model that included all predictors and interaction terms, the stepwise procedure evaluated the contribution of each predictor by assessing changes in AIC and deviance. Predictors that did not significantly improve the model were removed, and those that did were retained. This process continued until the model included only predictors that contributed significantly to explaining the variability in the outcome, as shown in the final equation.
(2)$$\log\left(E\left[Y_{i}\right]\right)=\beta_{0}+\beta_{\mathrm{gender}}{\mathrm{Gender}}_{i}+\sum_{j=1,j\neq j^{\prime }}^{k}\beta_{j}PC_{\mathrm{ij}}+\sum_{j=1,j\neq j^{\prime }}^{k}\sum_{l=1}^{m}\gamma_{jl}\left(PC_{\mathrm{ij}}\times \mathrm{Pat}h_{i}\right)$$

All statistical analyses were run in R (R Core Team, 2023) using the following libraries: haven, dplyr, FactoMineR, factoextra, glmnet, and car. Significance was set at 5%.

## 3. Results

### 3.1. Sample Characteristics

Appendix A presents the characteristics of the sample, which includes 226 patients stratified into the three risk paths, as follows: green (n = 35, 15.5%), yellow (n = 128, 56.6%), and red (n = 63, 27.9%).

In terms of pre-operative information, the distribution of sex differed significantly across the paths (*p* < 0.001). In the green path, 25.7% were male, and 74.3% were female. In the yellow path, 67.2% were male and 32.8% were female. In the red path, 57.1% were male and 42.9% were female. The mean age was 65.96 years (SD = 14.97) and did not differ significantly across the paths (*p* = 0.226). BMI categories varied significantly across paths (*p* = 0.002). In the green path, 5.7% were underweight, 80% had a normal BMI, 11.4% were overweight, and 2.9% were obese. In the yellow path, 8.6% were underweight, 55.5% had a normal BMI, 28.1% were overweight, and 7.8% were obese. In the red path, 9.5% were underweight, 34.9% had a normal BMI, 38.1% were overweight, and 17.5% were obese.

Regarding comorbidities, 14.6% of the overall sample had diabetes, with no significant difference across paths (*p* = 0.219). IBD was present in 7.1% of the patients, with no significant difference across paths (*p* = 0.12). Hematologic disorders were observed in 8.8% of the patients, with no significant difference across paths (*p* = 0.256). Cardiovascular disorders were present in 39.8% of the patients, with no significant difference across paths (*p* = 0.747). Psoriasis affected 0.4% of the patients, with no significant difference across paths (*p* = 0.273). Atopic dermatitis was reported in 2.2% of the patients, with no significant difference across paths (*p* = 0.309). Allergic dermatitis was found in 8% of the patients, with no significant difference across paths (*p* = 0.075).

In terms of previous treatments, 21.7% had received neoadjuvant chemotherapy, with no significant difference across paths (*p* = 0.343). Radiotherapy was administered to 14.6% of the patients, with significant differences across paths (*p* = 0.02).

The type of stoma planned showed significant differences across paths (*p* = 0.028). Colostomy was planned for 37.2% of the sample, ileostomy for 40.3%, and urostomy for 22.6%. Additionally, 81.4% had a scheduled stoma site implemented, with significant differences across paths (*p* < 0.001).

Job types varied among patients, with 9.3% having sedentary jobs, 31.9% having semi-sedentary jobs, 20.8% having non-sedentary jobs, and 38.1% not working. There was no significant difference across paths regarding job types (*p* = 0.242). Regarding sports activity, 55.3% never engaged in sports activities, 27.4% rarely engaged in sports activities, 12.4% often engaged in sports activities, and 4.9% always engaged in sports activities, with no significant difference across paths (*p* = 0.517).

Baseline risks varied significantly across paths. The median risk of dermatitis, itchiness, infections, and ulcers were significantly different across the paths (*p* < 0.001 for dermatitis, itchiness, and ulcers; *p* = 0.016 for infections). The median average risk of complications also varied significantly across paths (*p* < 0.001).

In the post-operative phase (15 days) at the Dermamecum baseline, 81% had a scheduled stoma site implemented, with no significant difference across paths (*p* = 0.41). The type of diet varied significantly across paths (*p* < 0.001). The abdomen profile also varied significantly across paths (*p* = 0.03), as did the physical form of the stoma post-surgery (*p* = 0.05).

Regarding post-surgical herniation and prolapse complications, 3.1% experienced an incisional hernia, 2.2% experienced a prolapse, and 0.4% experienced a hernia, with no significant difference across paths for any of these complications (*p* = 0.213, *p* = 0.141, and *p* = 0.681, respectively).

Regarding the devices used, 47.4% used a single-piece device, and 52.6% used a two-piece device, with no significant difference across paths (*p* = 0.964). The type of device baseplate used varied significantly across paths (*p* < 0.001), with 41.8% using a flat baseplate, 55.6% using a traditional convex baseplate, and 2.6% using a deep convex baseplate.

The baseplate material also showed significant differences across paths (*p* = 0.005). Hydrocolloid baseplates were used by 75.5% of patients, and hydrocolloid plus non-woven textile baseplates were used by 24.5% of patients.

Caregiver support levels did not differ significantly across paths (*p* = 0.202). Full caregiver support was provided to 33.2% of patients, partial support to 23.5%, supervisory support to 15.8%, and no support to 27.6% of patients.

Regarding ostomy care products, 12.8% used paste with alcohol, 29.2% used alcohol-free paste, 45.6% used protective film, and 75.2% used remover. The use of protective film (*p* < 0.001) and remover (*p* < 0.001) showed significant differences across paths. Other products, such as belts (7.1%), rings (4.4%), powder (14.2%), and extenders (2.7%), did not show significant differences across paths (*p* = 0.267, *p* = 0.3, *p* = 0.702, and *p* = 0.087, respectively).

In terms of hygiene, 46.9% used neutral pH soap, 27.1% used acidic pH soap, and 0.4% used basic pH soap. The use of acidic pH soap showed significant differences across paths (*p* < 0.001), whereas the use of neutral pH soap (*p* = 0.348) and basic pH soap (*p* = 0.681) did not. Non-woven gauze was used by 50.4% of patients, with significant differences across paths (*p* < 0.001). Wet wipes were used by 17.7% of patients, with no significant difference across paths (*p* = 0.094).

### 3.2. Primary Outcome: PSCs 30 Days after Surgery

A significant difference in the prevention of early complications was noted across the risk paths (*p* < 0.001). Specifically, 65.7% of patients in the green path, 68.8% in the yellow path, and 34.9% in the red path had their early complications prevented based on baseline risks.

When examining the observed complications within the 30-day post-operative period, 5.9% of the overall sample experienced complications. There was no significant difference in the observed complication rates across the different paths (*p* = 0.685). In the green path, 5.7% of patients had complications, compared with 4.7% in the yellow path and 7.9% in the red path.

### 3.3. MCA of the Theoretical Predictors

The scree plot (Figure 1) illustrates the percentage of variance explained by each dimension. The first ten dimensions explain a cumulative variance of 64.2%, with the first two explaining 11% and 9.2%, respectively.

Figure 2 presents the MCA factor map, displaying individual the distribution of data points (patients) across the first two dimensions, which contributes to explaining variance more than other dimensions (biplot of dimension 1 and dimension 2). The data points are colour-coded according to their respective risk stratification paths: green, yellow, and red. The ellipses represent the concentration of data points within each path, indicating each group’s central tendency and spread. The risk stratification paths (green, yellow, red) show distinct separations in the factor map, validating the effectiveness of the stratification based on performed assessments in the study. Each path forms a distinct cluster, indicating that patients within each path share similar characteristics, as captured by the first two dimensions of the MCA. The ellipses around each cluster represent the central tendency and spread of the patients within each path. The separation of ellipses indicates minimal overlap between the groups, suggesting successful stratification and distinct patient profiles within each path. Patients in the green path are more tightly clustered, indicating less variability within this group than in the yellow and red paths.

### 3.4. Predictors of PSCs 30 Days after Surgery

The initial Poisson regression model, which had an AIC of 90,645 and deviance of 32,675, included 5 principal components, gender, and the risk stratification paths (yellow and red paths, with the green path as the reference). Interaction terms between these principal components and the risk stratification paths, as well as gender and the risk stratification paths, were also considered. However, none of these predictors or interaction terms showed significant associations with PSCs at 30 days post-surgery.

The stepwise Poisson regression model, which had a lower AIC of 62,684 and a deviance of 42,684, indicated a more parsimonious fit. In this model, the intercept was significant (IRR = 0.024, 95% CI: 0.006–0.097, *p* < 0.001), suggesting a baseline risk for PSCs. Being male significantly increased the likelihood of developing PSCs within 30 days post-surgery, with an IRR = 4.033 (95% CI: 1.014–20.000, *p* = 0.048). None of these interaction terms reached statistical significance. Details of the models are shown in Table 2.

## 4. Discussion

Within the exploratory nature of an interim analysis, this study highlights several key findings, providing an initial understanding of the stratified approach to managing PSCs. The employed structured educational intervention, rooted in patient risk stratification [7], appears to have a promising impact on preventing early-onset PSCs. This study’s primary novelty lies in applying a structured educational approach (Dermamecum) that categorizes patients into risk paths (green, yellow, red) based on their pre-operative and post-operative characteristics. This stratification aims to tailor interventions more precisely to individual patient needs, a method supported by the literature emphasizing the importance of personalized care in ostomy management [22].

A major result of this study is the significantly lower incidence of PSCs compared with the median rates reported in the literature [1]. The observed complications within 30 days after surgery did not significantly differ across the paths, with 5.9% of the overall sample experiencing complications, indicating that the Dermamecum seems to prevent PSCs across every risk profile group. Previous studies have indicated a median PSC rate of 35%, with some variations depending on specific patient populations and study conditions [1,3]. In contrast, our study observed PSC rates equivalent to the lower bound of these estimates (6%), which strongly supports the effectiveness of the Dermamecum approach. This substantial reduction in PSCs corroborates the hypothesis that structured educational intervention could significantly improve patient outcomes when tailored to individual risk profiles [23,24,25].

The distribution of patient characteristics across the three risk paths revealed significant differences, particularly in gender, BMI categories, and planned stoma types. These differences validate the necessity for a tailored approach. For instance, the green path, characterized by a higher proportion of females and normal BMI patients, might inherently represent a lower-risk group [25,26]. In contrast, the yellow and red paths, with higher proportions of males and patients with higher BMI or obesity, highlight groups that might benefit more from targeted interventions. These results corroborate the discriminant validity of the risk assessment included in the Dermamecum, as well as the results derived from the MCA.

The MCA results provide a clear validation of the risk stratification [27,28]. The first two dimensions explain a significant portion of the variance, and the distinct clustering of patients within each path (as shown in the factor map) suggests that the stratification effectively captures the underlying risk profiles. The tighter clustering of the green path indicates lower variability within this group, while the broader spread of the yellow and red paths suggests more diverse risk factors.

The regression analysis further elucidates the complexity of predicting PSCs. It is important to note that the main aim of this study was not to identify predictors but to explore predictive behaviours and provide information to power the final study properly [29]. Using the components derived from the MCA to optimize the available power was a strategic choice [30]. Including all theoretical predictors in the regression models would have required a much larger sample size [7], which was beyond the scope of this interim analysis. By demonstrating the feasibility of this approach we showed that the model fit was adequate even when the only significant predictor after the stepwise selection was male gender, which holds intrinsic methodological novelties as well. The initial model, though comprehensive, did not identify significant predictors. In contrast, the stepwise model highlighted being male as a significant risk factor, with males having over 4 times the risk of developing PSCs within 30 days post-surgery. This finding aligns with previous studies indicating gender differences in wound healing and infection rates, which may necessitate gender-specific interventions in ostomy care [7,31].

The findings from this study have significant implications for specialized nurses in stoma care and interprofessional practice. The success of the Dermamecum protocol in reducing PSCs highlights the critical role of specialized nursing interventions tailored to patient risk profiles. Specialized nurses must be adept at implementing risk stratification-based educational approaches to enhance patient outcomes [23,24]. Moreover, the study underscores the importance of interprofessional collaboration, as managing PSCs effectively requires coordinated efforts among nurses, surgeons, urologists, dermatologists, and other healthcare professionals. This collaborative approach ensures comprehensive care, addressing various aspects of patient health and promoting better recovery and quality of life for stoma patients. Future research should continue to explore and refine this approach to further optimize stoma care, and further final powered studies are needed to corroborate this initial evidence [32,33,34].

This study has several limitations primarily related to the inherent constraints of conducting an interim study. First, the sample size, although adequate for preliminary insights, may not be large enough to detect all significant predictors of PSCs. This aspect limits the generalizability of the findings. Second, the short follow-up period of 30 days post-surgery restricts the ability to assess long-term outcomes and the sustained effectiveness of the Dermamecum protocol. Third, as this is an interim analysis, the findings should be interpreted with caution and validated with the final study results, which will incorporate a larger sample size and longer follow-up period to provide more robust conclusions. These limitations highlight the need for ongoing research to refine and optimize the structured educational approach in stoma care.

## 5. Conclusions

This interim study of the Dermamecum protocol demonstrates the potential effectiveness of a structured educational approach based on patient risk stratification in reducing early PSCs. The significantly lower incidence of PSCs compared with the median rates reported in the literature supports the efficacy of tailored interventions. The analysis highlights the importance of gender-specific strategies, with males showing a higher risk of PSCs. While the interim nature of this study imposes limitations, the findings provide a strong foundation for refining the final study design and optimizing future stoma care practices. Further research with larger samples and longer follow-up periods is necessary to validate these preliminary results and enhance the overall management of ostomy patients.

## Figures and Tables

**Figure 1 healthcare-12-01805-f001:**
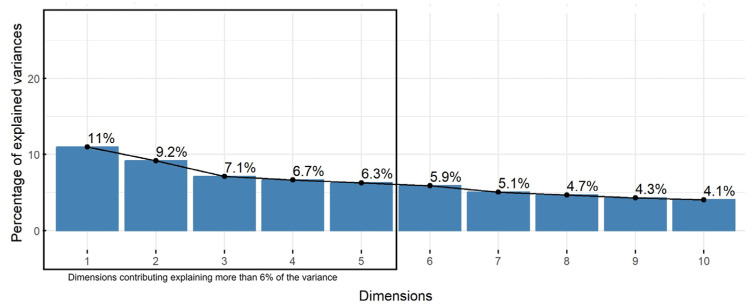
Scree plot.

**Figure 2 healthcare-12-01805-f002:**
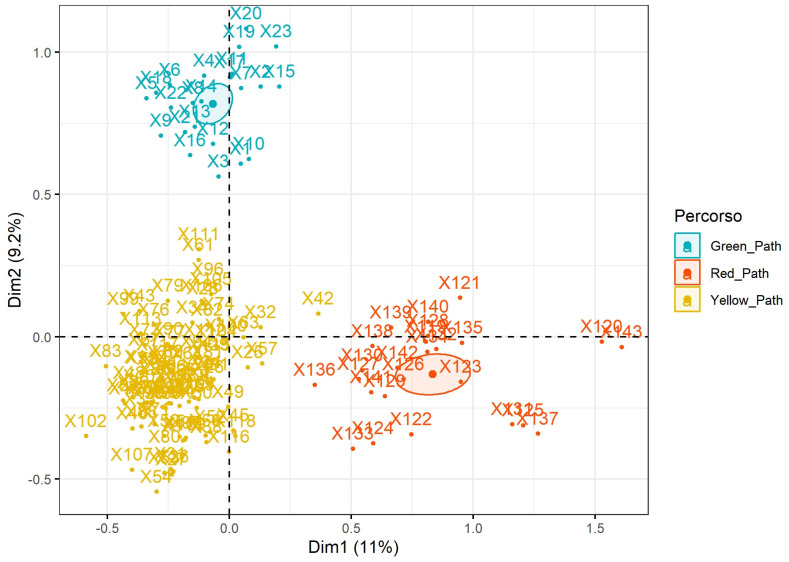
MCA factor map based on risk stratification paths.

**Table 1 healthcare-12-01805-t001:** Patient characteristics and predisposition to peristomal skin complications based on the study of Guerra et al. (2023) [7].

Complication Type	Predisposing Factors	Protective Factors
Dermatitis	Presence of psoriasis, chemotherapy treatment, use of one-piece ostomy devices, irregular stoma profile	Hydrocolloid plaques, low pH detergents, low-residue diet
Itchiness (Pruritis/Xerosis)	Overweight/obesity, male gender, use of deep convex plaques	Older age, sedentary lifestyle
Infections	Class 2 obesity or higher, frequent physical activity, hernias	Use of one-piece ostomy devices
Ulcerations	Underweight, non-autonomous status, inflammatory bowel disease, history of radiotherapy	Use of protective film, regular abdominal stoma profile

**Table 2 healthcare-12-01805-t002:** Poisson regression models.

Initial Poisson Regression Model (AIC = 90.645; Deviance = 32.675)				
Predictor	IRR	95% CI Lower	95% CI Upper	*p*
(Intercept)	0.00 × 10^0^	0.00 × 10^0^	0.00 × 10^0^	0.992
Dimension 1	1.14 × 10^−3^	0.00 × 10^0^	3.40 × 10	0.321
Dimension 2	3.12 × 10^−2^	0.00 × 10^0^	1.92 × 10^9^	0.470
Dimension 3	1.68 × 10^0^	2.0 × 10^−6^	1.25 × 10^11^	0.941
Dimension 4	1.37 × 10^2^	4.4 × 10^−5^	1.12 × 10^10^	0.498
Dimension 5	7.41 × 10^−1^	1.3 × 10^−4^	4.23 × 10^2^	0.942
Gender (reference is male)	1.25 × 10^7^	6.7 × 10^−3^	2.34 × 10^17^	0.994
yellow path	3.10 × 10^6^	5.2 × 10^−4^	1.86 × 10^11^	0.994
red path	4.22 × 10^6^	2.0 × 10^−6^	1.45 × 10^13^	0.994
Interaction: Dim1 × yellow path	1.90 × 10^2^	0.00 × 10^0^	5.72 ×10^14^	0.481
Interaction: Dim1 × red path	2.99 × 10^3^	1.7 × 10^−2^	2.53 × 10^14^	0.266
Interaction: Dim2 × yellow path	1.09 × 10^2^	0.00 × 10^0^	3.65 × 10^14^	0.419
Interaction: Dim2 × red path	6.91 × 10	0.00 × 10^0^	2.19 × 10^14^	0.456
Interaction: Dim3 × yellow path	2.27 × 10^−1^	0.00 × 10^0^	3.53 × 10^5^	0.838
Interaction: Dim3 × red path	8.05 × 10^−2^	0.00 × 10^0^	2.06 × 10^2^	0.749
Interaction: Dim4 × yellow path	5.40 × 10^−3^	0.00 × 10^0^	3.42 × 10^5^	0.505
Interaction: Dim4 × red path	2.96 × 10^−3^	0.00 × 10^0^	1.11 × 10	0.449
Interaction: Dim5 × yellow path	2.68 × 10^−2^	0.00 × 10^0^	7.45 × 10^6^	0.525
Interaction: Dim5 × red path	4.99 × 10^−1^	7.0 × 10^−6^	3.77 × 10^3^	0.879
Interaction: Being male × yellow path	0.00 × 10^0^	0.00 × 10^0^	1.46 × 10^7^	0.994
Interaction: Being male × red path	0.00 × 10^0^	0.00 × 10^0^	5.29 × 10^11^	0.994
Stepwise Poisson Regression Model (AIC = 62.684; Deviance = 42.684)				
(Intercept)	0.024	0.006	9.75 × 10^−2^	<0.001
Being male	**4.033**	**1.014**	**2.00**	**0.048**
Interaction: Dim1 × yellow path	1.90 × 10^2^	0.000	5.72 × 10^14^	0.481
Interaction: Dim1 × red path	2.99 × 10^3^	0.017	2.53 × 10^14^	0.266
Interaction: Dim2 × yellow path	1.09 × 10^2^	0.000	3.65 × 10^14^	0.419
Interaction: Dim2 × red path	6.91 × 10	0.000	2.19 × 10^14^	0.456
Interaction: Dim3 × yellow path	0.227	0.000	3.53 × 10^5^	0.838
Interaction: Dim3 × red path	0.080	0.000	2.06 × 10^2^	0.749
Interaction: Dim4 × yellow path	0.005	0.000	3.42 × 10^5^	0.505
Interaction: Dim4 × red path	0.003	0.000	11.1	0.449
Interaction: Dim5 × yellow path	0.027	0.000	7.45 × 10^6^	0.525
Interaction: Dim5 × red path	0.499	0.000	3.77 × 10^3^	0.879
Interaction: Being male × yellow path	0.000	0.000	1.46 × 10^7^	0.994
Interaction: Being male × red path	0.000	0.000	5.29 × 10^11^	0.994

Note: Scientific notation was preferred in order to optimize space. The green path is not explicitly included as a predictor because it serves as the reference category. In categorical regression analysis, one category of each categorical variable is chosen as the reference group, and the other categories are compared against this reference group. Bold values are those showing *p* lower than 0.05.

## Data Availability

The data used in this study are available upon reasonable request to the corresponding author, subject to any ethical and legal constraints.

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
