# Peer review of "Exploring the Impact of a Structured Educational Approach on Peristomal Skin Complications: An Interim Analysis"

_healthcare, 2024, doi:10.3390/healthcare12181805_

Round 1

Reviewer 1 Report

Comments and Suggestions for Authors

Unfortunately, despite the development of surgical techniques, the intestinal stomas are still often performed especially in oncological patients, I have carefully read the examination. This is a very interesting, thoughtful and exemplary project.

However, I have a few questions, please indicate what factors predisposed to specific groups, I did not find the exact qualification to the group, medium and high risk in the text, it should be described in the text in accordance with whether it was the tool of the collected multifactorial analysis. The concept of the dermaceum is clear and clearly described does not raise any objections.

The methodology is understood and described correctly, there is no information on ethical aspects, it is a large and serious scientific presowing whether it was reported to the bioethics committee, if you should not properly describe the note about the Helsinki Declaration.Conclusion and implications from practice are correctly defined.

Thank you that I could get acquainted with this material, congratulations on the idea and work.

Author Response

Comment 1: Unfortunately, despite the development of surgical techniques, the intestinal stomas are still often performed especially in oncological patients, I have carefully read the examination. This is a very interesting, thoughtful and exemplary project.

Response 1: Thank you for having appreciated our study and the idea to focus in peristomal complications of patients with stomas.

Comment 2: However, I have a few questions, please indicate what factors predisposed to specific groups, I did not find the exact qualification to the group, medium and high risk in the text, it should be described in the text in accordance with whether it was the tool of the collected multifactorial analysis. The concept of the dermaceum is clear and clearly described does not raise any objections.

Response 2: Thank you for your insightful comments. We have revised the manuscript to clarify that the stratification into Green, Yellow, and Red risk groups is based on the calculated percentage risk of specific peristomal skin complications (dermatitis, itchiness, infections, and ulcers), as identified in Guerra et al. (2023). These individual risk percentages are averaged to determine the patient's overall risk, which then guides their stratification into the appropriate risk group. We have included a detailed explanation of this process in the manuscript, with further specifics provided in Appendix A. We hope this revision satisfactorily addresses your concern.

Comment 3: The methodology is understood and described correctly, there is no information on ethical aspects, it is a large and serious scientific presowing whether it was reported to the bioethics committee, if you should not properly describe the note about the Helsinki Declaration.Conclusion and implications from practice are correctly defined. Thank you that I could get acquainted with this material, congratulations on the idea and work.

Response 3: Thank you for your feedback regarding the ethical aspects of our study. We have expanded the ethical considerations statement by adding a specific subheading in the manuscript to more explicitly outline our adherence to the Declaration of Helsinki and to detail the ethical considerations.

Reviewer 2 Report

Comments and Suggestions for Authors

The study is well-designed and performed, providing valuable insights into reducing peristomal skin complications (PSCs) and improving patient outcomes. The findings are significant and contribute to PSCs patient care by specialized nurses and other healthcare professionals.

There are a few suggestions to be detailed in the study:

a)     The Dermamecum was described into three distinct paths. The so-called ‘structured educational approach’ needs to be explained more systematically.

b)     How do we differentiate educational approach, educational initiative, and standard education?

c)     Did the authors find any dropout patients besides the three limitations mentioned in the study? If yes, please elaborate further. 

Author Response

Comment 1: The study is well-designed and performed, providing valuable insights into reducing peristomal skin complications (PSCs) and improving patient outcomes. The findings are significant and contribute to PSCs patient care by specialized nurses and other healthcare professionals.

Response 1: Thank you a lot for your appreciation of our study.

Comment 2: There are a few suggestions to be detailed in the study:

  1. The Dermamecum was described into three distinct paths. The so-called ‘structured educational approach’ needs to be explained more systematically.

Response 2: Thank you for your valuable feedback and for highlighting the need to provide a more systematic explanation of the structured educational approach within the Dermamecum framework. We have revised the relevant section of the manuscript to offer a clearer and more detailed description of the educational approach. Specifically, we expanded on how the Dermamecum categorizes patients into the Green, Yellow, and Red paths, outlining the objectives and educational content tailored to each risk group. This revision ensures that the structured nature of the educational strategy is fully articulated, demonstrating how the approach is designed to promote effective self-care, timely intervention, and improved patient outcomes. We appreciate your guidance and hope that these revisions meet your expectations.

Comment 3: b)     How do we differentiate educational approach, educational initiative, and standard education?

Response 3: Thank you for your insightful comment regarding the need to differentiate between "educational approach," "educational initiative," and "standard education." We have clarified these terms in the manuscript (subheading 2.3) to reflect their distinct roles within our study. The "educational approach" refers to the overarching methodology of risk-based education (Dermamecum), the "educational initiative" pertains to specific programs or interventions implemented within this approach, and "standard education" refers to the general education provided as part of routine care. We hope this clarification meets your expectations and enhances the general understanding of our educational strategy.

Comment 4: c)     Did the authors find any dropout patients besides the three limitations mentioned in the study? If yes, please elaborate further. 

Response 4: We would like to clarify that there were no dropout patients during the 30-day follow-up period.

Reviewer 3 Report

Comments and Suggestions for Authors

report what each colour path indicate at the abstract

why stoma patients present such complications- report the pathophysiolohy link

give the breif terminology of  interim analysis

maybe a table presenting the sample characteristics would be easier for the readers

very good methods section. WEll done

report as a table in the discussion or results section the patient characteristics which predispose to complications such as gender etc. - make more obvious your results

Author Response

Comment 1: report what each colour path indicate at the abstract

Response 1: Thank you for your insightful comment regarding the need to clarify what each color path (Green, Yellow, Red) indicates within the Dermamecum protocol. In response, we have revised the abstract to include a brief description of what each path represents

Comment 2: why stoma patients present such complications- report the pathophysiolohy link

Response 2: Thank you for this comment. We revised the introduction to explain why stoma patients are prone to peristomal skin complications by describing the underlying pathophysiological mechanisms.

Comment 3: give the breif terminology of  interim analysis

Response 3: We agree that defining interim analysis may help readers to understand the study. We added this definition in the design.

Comment 4: maybe a table presenting the sample characteristics would be easier for the readers

Response 4: We appreciate your feedback on making the information more accessible to readers. To provide comprehensive details, we have included all relevant sample characteristics in Appendix A of the manuscript. This appendix contains a thorough breakdown of the demographic, clinical, and post-operative characteristics of the study population.

Comment 5: very good methods section. WEll done

Response 5: Thank you.

Comment 6: report as a table in the discussion or results section the patient characteristics which predispose to complications such as gender etc. - make more obvious your results

Response 6: Thank you for your excellent suggestion to include a table summarizing the patient characteristics that predispose to peristomal skin complications. We agree that this addition would enhance the clarity and accessibility of our results. After careful consideration, we concluded that the most suitable part of the manuscript to incorporate this synoptic table is within the Methods section, specifically where we explain the stratification process. This placement allows us to directly connect the patient characteristics with the risk stratification, making the methodology clearer and more comprehensive for the readers. We appreciate your input and believe this addition strengthens the presentation of our findings.